# Zika Virus Pathogenesis: A Battle for Immune Evasion

**DOI:** 10.3390/vaccines9030294

**Published:** 2021-03-22

**Authors:** Judith Estévez-Herrera, Silvia Pérez-Yanes, Romina Cabrera-Rodríguez, Daniel Márquez-Arce, Rodrigo Trujillo-González, José-David Machado, Ricardo Madrid, Agustín Valenzuela-Fernández

**Affiliations:** 1Laboratorio de Inmunología Celular y Viral, Unidad de Farmacología, Sección de Medicina, Facultad de Medicina, Universidad de La Laguna (ULL), La Laguna, 38320 Tenerife, Spain; jesteveh@ull.edu.es (J.E.-H.); alu0101073996@ull.edu.es (S.P.-Y.); rcabrerr@ull.edu.es (R.C.-R.); alu0100540420@ull.edu.es (D.M.-A.); rotrujil@ull.edu.es (R.T.-G.); jdmacha@ull.edu.es (J.-D.M.); 2Unidad Virología y Microbiología del IUETSPC, Universidad de La Laguna (ULL), La Laguna, 38296 Tenerife, Spain; 3Departamento de Análisis Matemático, Facultad de Ciencias, Universidad de La Laguna (ULL), La Laguna, 38296 Tenerife, Spain; 4BioAssays SL. Campus de Cantoblanco, 28049 Madrid, Spain; rimadrid@ucm.es; 5Departmento de Genética, Fisiología y Microbiología, Facultad de Biología, UCM, 28040 Madrid, Spain

**Keywords:** Zika virus (ZIKV), immune evasion, infection, tissue propagation, congenital and neurological disorders

## Abstract

Zika virus (ZIKV) infection and its associated congenital and other neurological disorders, particularly microcephaly and other fetal developmental abnormalities, constitute a World Health Organization (WHO) Zika Virus Research Agenda within the WHO’s R&D Blueprint for Action to Prevent Epidemics, and continue to be a Public Health Emergency of International Concern (PHEIC) today. ZIKV pathogenicity is initiated by viral infection and propagation across multiple placental and fetal tissue barriers, and is critically strengthened by subverting host immunity. ZIKV immune evasion involves viral non-structural proteins, genomic and non-coding RNA and microRNA (miRNA) to modulate interferon (IFN) signaling and production, interfering with intracellular signal pathways and autophagy, and promoting cellular environment changes together with secretion of cellular components to escape innate and adaptive immunity and further infect privileged immune organs/tissues such as the placenta and eyes. This review includes a description of recent advances in the understanding of the mechanisms underlying ZIKV immune modulation and evasion that strongly condition viral pathogenesis, which would certainly contribute to the development of anti-ZIKV strategies, drugs, and vaccines.

## 1. Introduction

Between 2011 and 2018, World Health Organization (WHO) identified, alerted and tracked more than 1400 epidemic events in 172 countries due to emerging viruses, such as influenza, Severe Acute Respiratory Syndrome (SARS), Middle East Respiratory Syndrome (MERS), Ebola, Yellow Fever, Zika, and others [1]. These epidemic diseases are a prelude of the new global era the world is facing, and hallmark of high-impact, fast-spreading outbreaks that are increasingly difficult to manage, as is occurring nowadays with the new worldwide infection by the emerged SARS Coronavirus 2 (SARS-CoV-2) responsible for the ongoing coronavirus disease 2019 (COVID-19) pandemic [2]. Remarkably, in the last five years, WHO has declared three global health emergencies associated with emerging viruses: SARS-CoV-2 in 2020 [2], Ebola virus (EBOV) in 2014 [3,4,5,6] and Zika virus (ZIKV) in 2016 [7]. These threats were the first warnings of the socio-economic impact of emerging viruses, as the world is now experiencing with the SARS-CoV-2 pandemic which has become a global crisis [8]. The impact of the ZIKV pandemic put two of the biggest world sports events at risk of cancellation, both in Brazil, the Olympic Games (2016) and the World Soccer Championship (2018) [9,10].

In the context of ZIKV, WHO called Zika an “extraordinary event that needed a coordinated response, constituting a public health emergency of international concern (PHEIC)” [7], due to the description of a large outbreak of rash illness [11,12,13], with short-term and low-grade fever [13], not in all cases, and a cluster of microcephaly in newborns to infected mothers [14,15,16,17,18,19,20,21,22] together with neurological abnormalities and the Guillain-Barré syndrome (GBS) in Brazil [11,23,24,25,26,27,28,29,30]. In fact, ZIKV infection carries the risk of adverse pregnancy outcomes including increased risk of preterm birth, fetal death and stillbirth, and congenital malformations collectively characterized as congenital Zika syndrome (CZS), including the above-mentioned microcephaly, abnormal brain development, limb contractures, eye abnormalities, brain calcifications, and other neurologic manifestations [17,31,32,33,34,35]. ZIKV has been found in the cerebrospinal fluid (CSF) and brain of adults infected by the virus who manifested neurological disorders [24,36,37,38,39]. This flavivirus causes harmful effects in the adult brain, such as GBS [36,38,40,41,42], encephalitis [36,41,42,43], meningoencephalitis [24,44], acute myelitis [36,42,45] and encephalomyelitis [36,39,41,46,47], among sensory polyneuropathy [48] and other neurological complications [49,50]. After Brazil, other ZIKV outbreaks and rapid transmission were subsequently reported throughout the Americas and Africa, as well as other regions of the world [12,23,51,52,53,54,55,56,57,58,59,60,61,62,63,64,65]. Taken together, these data prompted the scientific community to retrospectively seek a potential association between ZIKV and reported cases of microcephaly and also GBS in French Polynesia (2013–2014) [66,67,68], providing a correlation of the risk of microcephaly, congenital malformations and fatal cases of fetuses and neonates from ZIKV infected mothers [69,70,71].

ZIKV is a mosquito-borne, positive-sense single-stranded RNA virus belonging to the family *Flaviviridae* (genus *Flavivirus*) [72,73,74,75,76,77,78,79,80,81,82,83]. ZIKV is further classified by homology to the Spondweni virus (SPONV) in the Spondweni viral clade or serogroup [73,84,85], both viruses were first characterized in Africa in 1947 and 1952 [82,84], respectively. ZIKV was isolated from the serum of a pyrexial rhesus monkey caged in the canopy of the Zika Forest in Uganda [82], and discovered during the study of the vector responsible for the cycle of sylvan Yellow fever virus (YFV) in Uganda [85]. The first confirmed human infections by ZIKV occurred in Nigeria (1954) [86], further cases were reported in Uganda (1962–63) [54] and outside Africa in central Java, Indonesia (1977) [87]. ZIKV presents three major phylogenetic related East African, West African and Asian/American lineages [88,89,90,91,92], constituting a single serotype [93]. These ZIKV linages are thought to have emerged from East Africa in the late 1800s or early 1900s [88,94], with the Asian lineage being responsible for all ZIKV outbreaks in the Pacific and the Americas [23,95,96,97,98,99,100]. 

ZIKV is principally transmitted to humans by infected *Aedes aegypti* and *Aedes albopictus* mosquitoes [100,101,102,103]. An increase of *Aedes* mosquito populations has been observed in tropical developing countries [104]. Thus, infected mosquitoes, highly populated areas, and global commercial and tourist activities together with modern transportation constitute an efficient scenario to spread infected mosquitoes and viruses such as ZIKV around the world [105], as reported with Dengue [104] and as occurred with epidemic ZIKV episodes in 2007 [95], 2013 [96,106] and 2014–2015 [107,108,109], declared as a PHEIC by WHO [7]. The scientific community efforts have rapidly increased knowledge about this virus. However, understanding the complexity of ZIKV infection, transmission and pathogenesis remains an urgent challenge. Hence, it is crucial to study competent vectors and natural reservoirs for ZIKV, its viral genetic diversity and flavivirus coinfection [110,111,112,113], as well as potential cross-immune reactivity (i.e., challenging the immune diagnosis) and immune enhancement of infection (i.e., antibody-dependent enhancement (ADE)) [114,115,116,117,118,119,120], fortunately not yet observed in humans [121], together with environmental factors that may suddenly trigger ZIKV epidemic expansion and a worse viral pathogenesis [12,30,55,122,123,124].

Due to the enormous challenge to develop a ZIKV vaccine [125,126,127,128,129], it is still not possible to immunize against ZIKV infection and associated pathology. Furthermore, the understanding of the immune events underlying ZIKV infection, transmission and pathology is key for meeting this ZIKV challenge. Based on the above, the present article aims to review the mechanisms of ZIKV immune evasion related to viral non-structural (NS) proteins, genomic and non-coding viral RNA, as well as microRNA (miRNA) generated during ZIKV infection to modulate cellular environment, in order to escape immunity and cause ZIKV complex pathology.

## 2. Mechanisms of ZIKV Immune Evasion

Apart from the CZS that covers the pathogenic events associated with maternal-fetal ZIKV transmission and the above summarized GBS [130], primary ZIKV infection is generally asymptomatic or mild in adults [131,132,133,134,135]. Factors determining asymptomatic or mild ZIKV infection, and severe manifestations, as well as chronic sequelae, are still to be determined. Understanding the importance of being infected by a particular ZIKV lineage, the influence of comorbidities and previous flavivirus infections (i.e., Dengue virus (DENV)), as well as viral load and the molecular mechanisms underlying severe infection, such as host genetic susceptibility to infection, immunosuppression and/or failure of the innate immunity are key to improving the overall knowledge of the complex ZIKV disease, diagnosis, prophylaxis and treatment [136,137,138,139,140,141,142,143,144,145,146,147].

The interplay between ZIKV and immune responses is initiated once the virus invades different cells, tissues and organs from the early infection. In infected individuals and non-human primates, the virus is fast cleared from blood, but persisting in saliva, urine, semen, breast milk and the central nervous system (CNS) for months [12,148,149,150,151]. Several in vitro and ex vivo studies indicate that ZIKV replicates in human endothelial and epithelial cells [152], peripheral blood mononuclear cells (PBMCs) [152], astrocyte and microglial cells [153,154], different placenta cells, such as trophoblasts [155], Hofbauer cells in chorionic villi and amniotic epithelial cells [156], as well as fibroblasts (placental, uterine, pulmonary) [157]. Moreover, ZIKV has a broad cell tropism in vitro, infecting human skin cells (i.e., dermal fibroblasts and epidermal keratinocytes), human myeloid cells (i.e., dendritic cells (DCs) and macrophages), and human progenitor cells of neuronal, placental and testicular origin (reviewed in [158,159]).

It is important to understand how ZIKV interacts with these cells and tissues, as well as with the host immune system to cause severe disease. The following sections describe the mechanisms underlying interferon (IFN) immune response against flavivirus, and the available knowledge concerning the events triggered by the ZIKV genome and proteins to escape or neutralize the antiviral IFN functions to infect and induce pathogenesis in immune privileged organs such as the brain and eye [140,160,161,162].

### 2.1. Antiviral IFN and Associated Signals

Host immune defense against ZIKV implies: (1) efficient recognition of the virus as the first line of defense (see Figure 1 for a schematic representation of ZIKV components that could be recognized by host defenses); (2) the induction of the IFN cascade intermediates and related signals, as a central mediator that allows antiviral action, inhibiting viral replication, and (3) initiation of a specific adaptive immune response [158,159,163,164,165,166], as occurs with other flaviviruses [159,167]. Nevertheless, this scheme is partially blurred by the action of the components of the virus. ZIKV responds to the assault by the infected cell by (1) mounting multiple camouflage strategies evading free genomic RNA recognition in cytosol, (2) avoiding the activation of multiple interactions of IFN cascades and finally (3) turning off the signaling that serves as a warning signal of infection.

The positive-sense single-stranded RNA (+ssRNA) genome of ZIKV is approximately 11 Kb in size and codes one large polyprotein that is processed by viral and host proteases into three structural (C, prM/M, and E) and seven non-structural (NS) proteins (NS1, NS2A, NS2B, NS3, NS4A, NS4B and NS5) [73,168,169] (Figure 1), as reported for other flaviviruses [170,171]. In infected cells, all these proteins are produced following genome replication and translation in the cytoplasm, where the structural proteins together with some NS proteins drive virion assembly at endoplasmic reticulum (ER) membranes and subsequent egress or new viral particles [172]. Furthermore, the enzymatic and protective battery of NS proteins are important for viral polypeptide processing, replication and modulation of host responses, which are needed for viral success [172]. Evolution studies of the ZIKV polyproteins suggest that NS maintains a high grade of conservation and presents several sites that mediate interaction with the host immune system [173] or that may regulate viral RNA synthesis [173,174]. Many of these functional sites have evolved to act against similar factors and functions making the host immune system antagonism redundant but adequately efficient, due to the need to competently escape the immune surveillance and to establish infection in the host [173,174]. Variability at these sites may modulate ZIKV infection capacity, optimization of immune evasion and pathogenicity, and this could be associated with the infection severity degree of the different viral phenotypes or lineages [140,142,175,176,177,178].

Once inside the target cell, as occurs with other RNA virus, the delivered +ssRNA genome of ZIKV could be recognized by pattern recognition receptors (PRRs), such as melanoma differentiation-associated gene 5 (MDA5) and retinoic acid-inducible gene 1 (RIG-I). This recognition activates PRRs synergistically and the subsequent signal pathway leads to expression of type I IFNs and other antiviral genes mediated by IFN regulatory factors (IRFs) [166,179,180], in general, accompanied with production of pro-inflammatory cytokines, in a nuclear factor kappa-light-chain-enhancer of activated B cells (NF-κB) dependent manner [181]. Furthermore, the active MDA5/RIG-I pathway joins several key factors, such as the mitochondrial antiviral signaling protein (MAVS), inhibitor of kappa-B kinase epsilon (IKKε), TANK-binding kinase 1 (TBK1), and IRF3 and IRF7, together playing a pivotal role in modulating the expression of type I IFN genes [180]. Type I IFNs continue the antiviral cascade in an autocrine and paracrine manner, by activation of type-I IFN receptors (IFNARs) and the crucial Janus kinase/signal transducers and activators of transcription (JAK/STAT (STAT1 and STAT2)) pathway in target cells. The activation of this axis evokes the second round of upregulation of antiviral genes which implies IFN-stimulated response elements (ISREs) that trigger the expression of IFN-stimulated genes (ISGs) to restrict ZIKV infection as a key action of the innate immune response [182,183,184]. These antiviral IFN-triggered ISGs exert diverse and different antiviral functions, such as inhibition of viral entry and viral protein synthesis, clearance of both viral protein and genome and inhibition of viral replication and release [183,185], as reported for ISG15 that limits ZIKV infection [186]. In addition, the antiviral responses failed during ZIKV infection in ISG15^-/-^ mice, with lower expression of RIG-I and IFN alpha-inducible protein 6 (IFI6) related ISGs and increased severity of retinal lesions [186]. Moreover, over-expression of IFI6 restricts ZIKV replication and prevents ZIKV-mediated cell death [187], as similarly reported for the IFN-inducible factor 16 (IFI16) [188]. However, ISG15 has also recently been reported to be involved in promoting ZIKV infection by regulating JAK/STAT and ISGylation (the term that stands for protein conjugation with ISG15 [189]) pathways [190].

IFNs are produced after ZIKV entry into target cells by recognition of the viral +ssRNA genome into the endosomal compartments by the membrane associated Toll-like receptors (TLR)-3 or -7 [191,192,193], as further confirmed by specific, small interference RNA (siRNA) knock-down of TLR-3 in human skin cells that strongly favors ZIKV infection [180]. ZIKV infection could also promote host mitochondrial DNA (mitoDNA) release (i.e., mitochondrial failure) [194] that is sensed by cyclic GMP-AMP synthase (cGAS) and signals through the ER associated intermediate stimulator of IFN genes (STING) [195]. Through these signals, as reported for DENV (reviewed in [196]), ZIKV may further activate TBK1 and IKKε which in turn phosphorylate IRF-3/7 [185], and IKKε activates NF-κB [197]. Finally, IRF3 and NF-κB act as transcription factors to promote expression of type-I IFNβ and some antiviral or proinflammatory genes [197].

Furthermore, paracrine IFN signaling in non-infected cells renders these cells refractory to viral infection. All type I IFNs are able to signal through the type-I IFNAR that is composed of two heterodimeric subunits (IFNAR1 and IFNAR2) which are generally widely distributed throughout the body (reviewed in [198,199]). This canonical type I IFN signal induces a plethora of ISREs which drive ISGs expression to establish a cellular antiviral state ([200,201,202,203]) (Figure 2). The protective role of IFN-I against ZIKV has been demonstrated in IFN-I signaling-deficient mice which are highly susceptible to viral infection [204]. Type I and II IFN-pretreated primary skin fibroblasts showed a non-permissive status against ZIKV, observing a strong and dose-dependent inhibition of viral replication [180]. In a different work, IFN-β shows moderated anti-ZIKV activity [205]. However, elevated secretion of IFN-β, detected in human lung epithelial cells during ZIKV infection appears to be responsible for preventing virus-mediated cell death by apoptosis [205]. Concerning the protective role of ISGs against ZIKV infection, it has been reported that placental cells could resist viral infection due to actions of IFN-λ (type III IFN) [206]. Likewise, it seems that small membrane-associated IFN-inducible transmembrane proteins (IFITMs), particularly IFITM1 and 3, could inhibit ZIKV infection early in the viral life cycle [207]. In this work, results point to the possibility that IFITM3 could also prevent cell death mediated by ZIKV infection [207].

In this scenario, ZIKV replication appears to be either impaired in IFN-α- or IFN-β-pre-treated primary skin fibroblasts [180], which are key IFNs for autocrine and paracrine antiviral effects [208], or promoted by anti-IFN-α/β receptor (IFNAR2) antibodies (Abs) in human infected DCs [209], or in mice lacking IFNAR1 or IRF3, 5 and 7 which developed more severe neurological disease compared to wild-type [210].

Therefore, ZIKV needs to neutralize and evade these type I IFN antiviral responses to infect and persist. In order to do this, the virus carries several components that are able to antagonize IFN antiviral functions, such as the structural E protein, NS proteins, genomic and non-coding viral RNA which are summarized in the sections below.

### 2.2. ZIKV Structural E Protein Inhibits IFN Production

In viral particles, the ZIKV E surface glycoprotein is responsible for the binding of the virion to different receptors at target cells, thereby determining viral tropism, infection and disease (reviewed in [168]). Furthermore, the ZIKV E protein elicits a strong humoral immune response in infected individuals that has positioned it as an ideal candidate for the development of vaccines. However, there is a disparity between the highly immunogenic nature of ZIKV E and the poor neutralizing capacity of the polyclonal Ab response induced against it [211].

A plethora of host receptors has been reported to mediate in ZIKV E-dependent attachment and/or entry and infection, such as heparan sulfate glycosaminoglycans (HS-GAGs) [212], the C-type (calcium dependent) lectin, dendritic cell-specific intercellular adhesion molecule-3 grabbing nonintegrin (DC-SIGN)/CD209 [180], and phosphatidylserine (PS) receptors of the TIM (T-cell immunoglobulin and mucin domain) and TAM (Tyro3, Axl, and Mer) families (reviewed in [168]). ZIKV indirectly binds to the Axl (Anexelekto; a Greek word that means uncontrolled) TAM kinase through Gas6 (growth arrest-specific 6) soluble factor, the natural ligand of Axl (i.e., both factors constituting an Axl-Gas6 complex). The functional role of this TAM kinase in ZIKV infection is mediated by the ability of Gas6 to bridge with protein S and the PS on the viral envelope [213,214,215,216]. These ZIKV-PS/Gas6-Axl interactions therefore mediate the attachment of the Zika virion to the host cell-surface [153,154,217].

It is important to note that activated Axl kinase suppresses type I IFN signaling [218]. ZIKV could therefore condition immune escape from the first steps of the viral life cycle, through activating Axl and impairing IFN production, then negatively acting on IFN-defenses [154,213]. Thus, the efficiency of ZIKV to target the different host receptors during the entry mechanism could compromise the efficiency of the IFN protection, which could explain the lack of an effective vaccine against ZIKV E, since this viral antigen may suppress IFN defenses [213,219,220,221,222].

### 2.3. ZIKV NS Proteins as Modulators of Antiviral IFN and Associated Signals

Besides their central roles in the viral lifecycle [223], NS proteins modulate innate responses altering IFN cascades and associated protective mechanisms (Figure 2). NS1 targets the IFN-associated cGAS-STING pathway by promoting cGAS processing by caspase-1. NS1 stabilizes caspase-1 by recruiting the deubiquitinase USP8 that deubiquitinates caspase-1 at K134 residue, increasing its stability and facilitating its action to cleave cGAS which results in reduced type I IFN signaling and enhanced ZIKV replication [195]. In addition, a natural, evolutionary NS1 mutation (bearing 188V residue) in the Asian ZIKV lineage has been described which facilitates its interaction with TBK1, diminishing TBK1 phosphorylation that results in the inhibition of IFN-β production in human cells, and viral replication enhancement in immunocompetent C57BL/6J mice [224]. Moreover, NS1 together with NS4B target type I IFN production by stabilizing NS2B-NS3 (NS2B3) during viral infection [225].

NS2A down-regulates the promoter activity of IFN-β by acting on several factors that are key for IFN-β activation [226]. Hence, NS2A suppresses RIG-I and MDA5, two pattern recognition receptors (PRRs) of the viral RNA genome for triggering IFN production and antiviral functions [166,179]. It is noteworthy that NS2A inhibits IFN-β induction by all forms of the IRFs (IRF3) [226], which act as IFN promoters in the JAK/STAT pathway [182]. Down-stream from IFN signaling, the JAK/STAT pathway promotes antiviral ISGs which target different steps of the viral cycle [182]. As regards NS2B, overexpression of the ZIKV NS2B3 precursor also interferes with the JAK/STAT pathway, as it is involved the viral helicase domain in interacting with JAK1 and driving it to the proteasome degradative route [225]. On the other hand, and confirming the NS2B3 interplay with the antiviral IFN function, it has been reported that type I IFN signaling promotes NS2B3 autophagic clearance in a STAT1-dependent manner, thus limiting ZIKV replication [225]. Moreover, NS2B3 helps viral replication by inhibiting protective RIG-I-like receptor (RLRs)-mediated cell-apoptosis in a JAK1-independent manner [225]. RLRs recognize cytosolic viral RNA [227], acting as sensors for ZIKV and other RNA viruses [167]. Moreover, NS2B3 is able to target the cGAS/STING pathway by cleaving STING, a degradative action that abrogates ISGs expression in response to cGAMP and neutralizes anti-ZIKV defenses [228].

NS3 protein also blocks RIG-I- and MDA5-mediated IFN antiviral actions, but acts on the 14–3-3 family of adaptor proteins [229] These factors complex with either RIG-I or MDA5 to translocate them to mitochondria where they promote antiviral IFN induction [230,231]. Thus, NS3 binds to 14–3-3 proteins preventing 14–3-3 factors from working together with RIG-I or MDA5 to recognize viral patterns and neutralize ZIKV infection [229]. Regardless of these multiple mechanisms for evasion of viral sensing, NS2B-NS3 also induces nucleoporins (NUPs) degradation and causes a marked inhibition of mature messenger RNAs (mRNAs) and nuclear proteins export to the cytoplasm [232], performing novel functions hijacking the protective role of nuclear components relevant in triggering immune response pathways, thereby inducing a favorable environment for viral replication

The ZIKV-NS4A protein performs an anti-IFN activity, reducing its production by blocking RIG-I and MDA5 signaling, also at the mitochondrial level [226,233]. NS4A competitively binds to the N-terminal caspase recruitment and activation domain (CARD) of the mitochondrial MAVS factor, abrogating MAVS interaction with activated RIG-I or MDA5, resulting in the inhibition of type-I IFN production. It has also been reported that NS4A inhibits the activities of both IFN-β and IRF3 reporters, blocking the type-I IFN response [234,235]. Furthermore, NS4B has also been observed interacting with TBK1 kinase, preventing its oligomerization and phosphorylation that are responsible for its activation and subsequent IFN production [225]. Besides, NS4B in cooperation with NS1 also inhibits type I IFN production by stabilizing NS2B3 during viral infection [225].

ZIKV NS5 is the most highly conserved protein encoded by ZIKV genome not only because of the requirement to maintain two enzymatic activities indispensable for virus replication, methyltransferase and RNA-dependent RNA polymerase, but also in its IFN-I antagonism [173,236,237,238,239,240].

NS5 targets each level of the IFN activation axis in host cells, particularly impairing genomic RNA sensing at its 5′ untranslated region (UTR) capping by RIG-I by repressing RIG-I polyubiquitination by means of the NS5-MTase (methyltransferase) domain but MTase function independent, thereby making RIG-I incompetent to activate IRF3 and therefore IFN-β production [241], or by acting as a barrier of IFN activation (Figure 1). Hence, NS5 abrogates IRF3 and NF-κB signaling [234], as it is able to interact with IRF3 avoiding its activation [224], or binding to IKKε, affecting IKKε protein levels and phosphorylation, which also finally results in IRF3 inactivation [242]. Furthermore, NS5 protein antagonizes IFN production by impairing the activation of TBK1 and, therefore, the IRF3 factor, a TBK1 substrate for phosphorylation [243]. In addition, NS5 competitively binds to the ubiquitin-like domain of TBK1, affecting its interaction with TRAF6 (TNF (tumor necrosis factor) receptor-associated factor 6). This complex is required for TBK1-mediated IRF3 phosphorylation and activation, thus interfering with type I and III IFN transcription [243].

Additionally, in IFN-induced cells, ZIKV NS5 expression results in proteasomal degradation of the IFN-regulated transcriptional activator STAT2 in humans but not in mice, and also affects STAT1 phosphorylation levels, suppressing INF-mediated JAK/STAT signal transduction [244,245], which show that ZIKV-induced disease takes advantage of NS5-promoted IFN deficiency. Moreover, despite the fact that the role that NS5 plays in the nucleus remains enigmatic, the latest works suggest that subcellular localization of NS5 is important for its function in innate immune suppression which provides new insight into ZIKV pathogenesis [239,246,247]. Hence, NS5 allows ZIKV evasion of the IFN-mediated innate immunity.

### 2.4. Viral Genomic and Subgenomic Rnas Inhibit IFN Signaling

The ZIKV genome contains 5′ and 3′UTRs (Figure 1), about 107 and 428 nucleotides long, respectively [73,248,249]. The RNA genome of ZIKV and other flavivirus are processed by cellular exoribonucleases, such as the host 5′-3′ exoribonuclease (XRN-1) to produce subgenomic flavivirus RNAs (sfRNAs) once inside the cell [250]. These incompletely degraded sfRNAs result from the stalling of the host 5′-3′ XRN1 enzyme or its homologous Pacman in insects on conserved RNA structures in the 3′UTR of the ZIKV RNA [250,251,252,253]. Furthermore, XRN1 is halted in its degradative action by two structured XRN-1-resistant RNA elements (xrRNAs) on ZIKV, xrRNA1 and xrRNA2 [254,255] which protect the remaining 3′UTR leading to the accumulation of sfRNAs during ZIKV infection. The ZIKV genome also presents an additional putative xrRNA3 formed by a structured RNA element known as dumbbell (DB1) [255,256].

As regards IFN modulation, ZIKV sfRNAs seem to suppress RIG-I- and MDA-5-mediated IFN induction, as reported by an experimental approach based on transient expression of the ZIKV-3′UTR together with transfection of the RIG-I or MDA-5 agonist [257]. Moreover, nucleotide deletions in the DB-1 structure of the ZIKV-3′UTR region, in a live-attenuated vaccine candidate, impair viral RNA synthesis and increase the sensitivity to type I IFN inhibition [258]. Despite all these reported data, the mechanism underlying the observed attenuation of type I IFN production by sfRNAs is not well understood [248].

However, it is conceivable that sfRNAs may be involved in ZIKV efficient infection and neuro-pathogenicity in vivo, since these events have been related to impairment of the IFN response in several mouse models of ZIKV infection [204,259,260,261,262] and in other scientific evidence obtained from patients [206,235,263,264]. In this regard but without targeting IFN production, recent studies using wild-type ZIKV point to the importance of sfRNAs for viral infection in both human cells and mosquitoes [265]. ZIKV-sfRNAs do not negatively affect IFN levels, since high IFN and proinflammatory cytokines levels were induced after infection of human cells with wild-type virus [265]. However, IFN levels were strongly diminished in cells infected by a recombinant ZIKV unable to generate sfRNAs [265]. Of note, efficient viral clearance was observed in cells infected with sfRNA-deficient ZIKV. Further studies are needed to clarify this apparent contradictory observation concerning the absence of sfRNA-mediated regulation of IFN immunity. However, these data correlated well with the strong induction of ISGs which occurred in these infected cells, thereby suggesting that the deficiency in sfRNA is associated with an uncontrolled IFN signaling which strongly activates ISGs to further clear ZIKV [265]. Therefore, sfRNAs are required for ZIKV propagation in human cells. Furthermore, results obtained with recombinant ZIKV indicated that the absence of sfRNAs results in inefficient viral infection and transmission in *Aedes aegypti* mosquitoes [265]. Similarly, a recent study has demonstrated the importance of the sfRNAs for productive infection and virus transmission in mosquitos by suppressing ZIKV infection-mediated apoptosis and assuring the ability of the virus to disseminate and reach saliva in the invertebrate vector [266].

On the other hand, viral genomic RNA (gRNA) double methylation during flavivirus infection appears to be key for avoiding gRNA rapid degradation and recognition by antiviral host factors, and for controlling the efficiency of mRNA translation and [248,267]. 2′-O cap methylation of the 5′UTR end mediates the evasion of flavivirus from the antiviral IFN-induced proteins with tetratricopeptide repeats (IFIT) response [268], thereby inhibiting their binding to viral gRNA [267]. Hence, the West Nile virus (WNV), DENV and Japanese encephalitis virus (JEV) mutants, deficient in 2′-O methylation activity, are unable to escape viral suppression by IFN and IFIT proteins [268,269,270]. In this respect, 2′-O methylation of the 5′-end of the ZIKV gRNA has also been reported [271,272], with the NS5 MTase activity itself being responsible for this modification [271,272,273]. Thus, 2′-O methylated 5′-cap gRNA mimics cellular RNAs, evading its recognition by antiviral host factors and following translation by the host machinery [272]. It is plausible, therefore, that 2′-O methylation of the ZIKV gRNA mediates immune escape, viral infection and pathology.

These observations have promoted the development of ZIKV and other flavivirus antiviral drugs [271,274,275,276] and vaccines based on deficiency in 2′-O methylation, in order to potentiate IFN and IFIT antiviral functions and attenuating flavivirus virulence [270,277,278,279,280].

### 2.5. MicroRNA, ZIKV Infection, Immunity and Pathogenicity

A microRNA (miRNA or miR (*consensus nomenclature reviewed in* [281])) is an evolutionarily conserved small non-coding RNA, 20–25 nucleotide-long, that performs a wide regulatory activity on several biological processes such as cellular immune responses against viruses in the case of viral miRNAs [282]. miRNAs modulate gene expression through base pairing of the miRNA seed sequence, mainly located within the 3′UTR to its target mRNA, which leads to either translational repression or mRNA cleavage, blocking target protein synthesis [283,284]. In the context of ZIKV infection and pathology a new field of research has emerged to find potential viral and cellular miRNAs that could regulate immune responses to favor the viral life cycle and thereby contributing to ZIKV complex pathology.

In infection experiments of human neuronal stem cells (hNSCs) with ZIKV MR766 and Paraiba strains, data obtained indicate that genes regulating stem cell survival, *NESTIN* (neuroectodermal stem cell marker) and *PAX6* (paired box 6), cell cycle and neurogenesis are repressed in a miRNA-dependent manner [285]. In this work, ZIKV infection upregulated miR-124–3p and let-7c which downregulated transferrin receptor (*TFRC*) and HMGA2 (high-mobility group AT-hook 2) mRNAs, respectively, thereby downregulating these target genes, including the established regulator of NSC renewal mRNA in hNSCs and mouse brain tissue [285]. Moreover, ZIKV infection upregulated miR-125a-5p together with miR-125a-3p, both showing repressive activity on MAVS which is central in the RIG-I and type I IFN response pathways of the innate immune system [286]. Additionally, ZIKV infection upregulated miR-7–5p and miR-320c, both miRNAs are able to target “*SIN3 transcription regulator homolog A* (*SIN3A*)” mRNA, which codes for a paired amphipathic helix protein that bears a STAT3-interacting repressor, essential for IFN-mediated antiviral response [287]. The authors proposed that this miRNA-mRNA interplay could be key for suppression of IFN signaling in ZIKV-infected hNSCs and suggest that these miRNAs could orchestrate the suppression of gene networks important for immunity, fostering neurodegeneration and microcephaly.

In astrocytes, ZIKV infection has been mainly related to the downregulation of several miRNAs with the unfolded protein response (UPR) pathway as a major downregulated target [288]. Apart from this, a downregulated key gene during the course of ZIKV infection is the *DICER1* transcripts, which may account for global downregulation of miRNAs. On the contrary, ZIKV infection upregulates some miRNAs, such as miR-431-5p, together with upregulation of *CHOP* (*C/EBP homologous protein*) and *GADD34* (*growth arrest and DNA damage-inducible 34*) genes that are autophagy associated factors [288]. Considering all these data, it seems that many flaviviruses facilitate their own viral replication and persistence by targeting the UPR pathway [288,289,290]. The alteration of this cellular quality control mechanism promotes the accumulation of misfolded proteins and triggers ER stress and autophagy [291]. Thus, ZIKV infection activates the expression of proteins related to the UPR pathway in mouse neural cells, such as IRE1 (inositol-requiring enzyme)-XBP1 (X-box binding protein 1) and ATF6 (activating transcription factor 6) [292], also observing the upregulation of several RNA transcripts of genes related to the autophagy function and pathway, such as *atf4*, *gadd34*, *chop*, and *edem-1* genes. A recent study in ZIKV-infected primary mouse neurons states that ZIKV-modulated miRNAs include miR-155, miR-203, miR-29a and miR-124-3p [293]. Furthermore, these ZIKV-modulated miRNAs seem to interact with mRNAs that regulate neurological development and neuroinflammatory responses. Thus, some of these miRNAs have been involved in the control of flavivirus infection, anti-viral immunity and brain injury [294,295,296,297,298,299]. Likewise, it has been reported that ZIKV modulates mRNA levels of enzymes involved in the generation of miRNAs, such as Drosha, Dicer, Ago2, Ago3 and DGCR8, during infection in liver, lung and kidney cells [300]. Additionally, virus-derived siRNAs (vsiRNAs) have been observed in ZIKV infected human neural progenitor cells (hNPCs), as well as in mosquito cells [301]. These vsiRNAs seem to exert an immune antiviral effect, since targeting key RNA interference (RNAi) machinery components significantly enhances ZIKV replication [301]. This antiviral action exerted by vsiRNA has been confirmed by using brain organoids treated with the RNAi enhancer enoxacin [302]. The increase of antiviral RNAis by enoxacin inhibits ZIKV-induced microcephalic phenotypes in brain organoids [301]. Therefore, the regulation of these RNA transcripts and proteins may account for the permissivity of these organs for ZIKV infection and damage.

On the other hand, several studies challenging human and mouse neuronal cells with ZIKV have not reported any change in the expression level of miRNAs expression in cells of the astrocyte lineage [303,304,305].

Apart from these discrepancies and the necessity of an accurate mechanistic characterization of ZIKV induction or repression of miRs, in target cells and tissues, it is conceivable that the regulation by ZIKV of the interplay between miR and mRNA transcripts of key genes for immune functions, cell differentiation and fate would be crucial to understand the events underlying viral immune escape and neuropathogenesis, during active ZIKV infection and transmission Therefore, these ZIKV related miRs are good candidates to develop therapeutics for the management of ZIKV neurological disease.

### 2.6. ZIKV Infection and Adaptive Immune Responses Understanding Cross-Immune Protection and Vaccine Challenges

The innate immune responses imply a rapid sensing of viruses to eliminate them, and ZIKV deploys different strategies to subvert these mechanisms, as presented above. Adaptive immunity against viruses provides a more finely tuned repertoire of recognition for these nonself-antigens (reviewed in [306,307,308,309]). The adaptive immune response is initiated by activation of naïve T cells through antigen presentation, where CD4^+^ and CD8^+^ T cells are important players in the control of viral infection, representing the memory cell repertoire of defense for future expositions to the virus together with B cells and neutralizing Abs (NAbs) (reviewed in [306,308,309,310,311]).

Some studies on the role of T cells in ZIKV infection indicate that CD8^+^ T cells are important for controlling the viral load (viremia) of ZIKV, as depletion of CD8+ T cells in mice resulted in enhanced virus replication [312], whereas CD4^+^ T cells presented a Th1 profile that produces the antiviral IFN-γ together with TNF-α, and interleukin 2 (IL-2) [312]. These data have been confirmed by other studies of primary ZIKV infection in animal models, where CD8^+^ T cells exert a key protective action against ZIKV [313,314]. CD8^+^ T cell protective responses against ZIKV have been reported to target structural proteins such as E, prM, and C [315], whereas CD8^+^ T cell responses promoted by DENV infection are predominantly against NS proteins, such as NS3, NS4B and NS5 [316,317,318].

Cross-reactive T cell-mediated protective action against ZIKV infection has gained interest since ZIKV affects regions where DENV used to be endemic [159,319,320], and the scientific community is concerned about potential reinfection events possibly causing acute or more severe ZIKV infection and pathogenesis [114,320,321]. Thus, characterization of this flaviviral T cell cross-protection is crucial for vaccine development and preparedness actions for future outbreaks [322,323,324,325]. In this regard, CD8+ T cell cross-response against ZIKV has been reported with CD8^+^ T cells generated during a previous DENV infection. In mice, DENV-elicited CD8^+^ T cells protect against a posterior challenge with ZIKV either in non-pregnant [326,327] or pregnant mice [328]. In addition, both immunization with ZIKV-specific or ZIKV/DENV cross-reactive peptides, followed by ZIKV challenge, elicited CD8+ T cell responses that were protective as depletion of CD8+ T cells resulted in increased ZIKV infection [326]. Similar data have been reported with DENV2. Hence, DENV2-promoted protective CD8^+^ T cells against ZIKV infection could recognize epitopes in different NS and structural ZIKV proteins, mainly NS3, NS5, prM and E [319]. This cross-immune protection has been corroborated in humans by a study in DENV-endemic countries (Sri Lanka and Nicaragua) and by comparing ZIKV cross-reactive T cells from non-infected individuals vs. individuals that have been previously exposed to DENV [315]. This study discovered an interesting observation relating the distinct specificity for ZIKV proteins and epitopes shown by protective CD8^+^ T cells generated in ZIKV infected individuals that have previously experienced a DENV infection compared with individuals without any previous reported DENV infection. Thus, CD8^+^ T cells from DENV-immune individuals showed a response to epitopes in ZIKV NS3 and NS5 proteins, whereas cells from DENV-naïve individuals targeted C, E, and prM structural proteins [315]. The predominance of DENV-elicited CD8^+^ T cell cross-immune reactions against ZIKV-NS proteins have also been observed in other human studies, mainly recognizing NS3 protein [329,330] with poor cross-reaction reported against structural proteins, such as ZIKV C protein [330]. Therefore, human pre-existing T cell responses against DENV have been found to recognize some peptide sequences in ZIKV proteins, which could be explained by the high degree of homology in sequence and structures (DENV and ZIKV polyprotein homology is about 55–56%) together with functional long-range interactions between the two flaviviruses [319,331]. These facts allow previous DENV infection to promote reactive T cells that cross-react and protect against ZIKV infection, as further reported by other studies in humans [332,333,334].

It has been reported that peptide immunization or ZIKV infection-elicited CD4^+^ T cell responses are necessary for protection against virus infection and further viral clearance in primary infection. However, the involvement of viral antigen specific CD4^+^ T cells in the control of ZIKV infection and disease is still controversial [335,336]. ZIKV-triggered CD4^+^ T cell mediated Ab response, but not CD8+ T cells, seemed to be responsible for viral clearance in an intravaginal infection model [336]. These data point to the relevance of CD4^+^ T cell immunity in blocking sexual transmission. Activation of CD4^+^ T cells and associated humoral immunity should therefore be considered as a challenge for vaccine designs that aim to interfere with ZIKV sexual transmission. Vaccine-mediated T cell protective recruitment is also key to achieving protection against ZIKV in the maternal-fetal interface during pregnancy. This is a real challenge to be accomplished by vaccines, since T cells mobilization and access to the maternal-fetal interface is restricted during pregnancy [337]. In this regard, attenuated ZIKV used as vaccine failed to elicit strong T cells responses during pregnancy in mice, when compared with vaccine efficiency at that level in non-pregnant mice. Humoral protection of this tissue in pregnant mice requires higher NAbs titers in order to protect and block vertical transmission of ZIKV [338]. Therefore, ZIKV could enter an immune privileged interface and tissues during pregnancy, therefore a vaccine should offer high potency in eliciting CD4^+^ T cell activation to cooperate with B cell to assure long-term associated humoral immunity, together with protective CD8^+^ T cells.

For the above reasons, the efforts to develop anti-ZIKV vaccines are focused on generating protective CD8^+^ and CD4^+^ T cells to control ZIKV infection and promote virus clearance. In this respect, an NS3-based vaccine has recently been reported to generate specific CD8+ T cell response protecting against ZIKV infection [339]. These vaccine-elicited polyfunctional CD8^+^ T cells are required to prevent death in lethally infected adult mice and fetal growth restriction in infected pregnant mice. Moreover, this NS3 vaccine design and strategy appears to be safe, since it failed to induce a NAb response [339], thereby avoiding serious concerns related to potential ADE that could increase the severity of a reinfection event [340] that in the flavivirus family could occur with a secondary exposure with a different serotype of the same virus or with heterologous flavivirus [340]. Although it has been reported that anti-DENV Abs (i.e., generated during DENV infection) could enhance ZIKV infection and disease severity in mice [117,341,342] as well as in human placental explants [343], there is not any evidence that clearly demonstrates ZIKV-ADE in humans [340,344]. This safety concern is always behind several anti-ZIKV vaccine designs and developments that are conceived to elicit NAb responses, and use prM and E (prM-E) proteins in the majority of the anti-ZIKV vaccine candidates developed so far [345]. Furthermore, a DNA vaccine expressing ZIKV NS1 protein protected against viremia and disease in BALB/c mice infected by ZIKV [346]. In this case, it seems that viremia is decreased by the NAbs generated by the NS1 DNA vaccine, with CD4^+^ and CD8^+^ T cells also being key for viral control, since this vaccine failed to protect against a lethal ZIKV challenge in *Ifnar1^−/−^* mice [346]. Therefore, and considering that ADE effect is not demonstrated in humans for ZIKV infection [340], this NS1-based vaccine design that elicits specific and efficient humoral and cellular immune responses may be a good candidate for anti-ZIKV vaccine in humans to elicit long-term protection. Additionally, the above mentioned NS3-based vaccine that activates protective CD8^+^ T cells against ZIKV in a highly susceptible mouse model with human-relevant T cell responses, prompted the suggestion that this vaccine might afford protection in vulnerable pregnant individuals [339], representing a promising candidate for a safe anti-ZIKV vaccine in humans and evading ADE-associated concerns [344]. Moreover, an adenovirus type 4 vector-based vaccine expressing ZIKV prM-E achieved protection against lethal ZIKV challenge in mice, without generation of anti-ZIKV NAbs [347], which could be also considered a safe vaccine candidate.

A vaccine to protect against ZIKV infection, transmission and pathogenesis is a challenge that requires deep molecular and cell/tissue knowledge on how the virus infects and persists through several cell types and organs, particularly, in immune privileged tissues where vaccines should induce IFN-mediated innate antiviral immunity, and recruit long-term memory cellular and humoral protective immunity.

### 2.7. ZIKV Invades and Infects the Fetal Brain and Eye: Evading Immune Responses in Immuno-Privileged Organs

ZIKV is able to persist for months in immune-privileged sites, such as testes that allows it to be sexually transmitted [348], eyes [160] and the CNS [140,162]. The infection of the placenta and fetus by ZIKV is proof of the innate and adaptive immune evasion of this virus in relevant cell types and tissues/organs which are related to the CZS and neuro-damage and disorders [130].

The interplay of ZIKV with these relevant tissues could disturb metabolism and the fetus-mother exchange of different important factors, such as nutrients, growth factors and gas (CO_2_/O_2_)-exchange in the placenta during pregnancy, and altering vasculature and nervous system development [349]. ZIKV is able to infect and replicate in different primary cell types isolated from placental chorionic villi and the chorioamniotic membranes (e.g., cytotrophoblast cells, Hofbauer cells, amniotic epithelial cells, and umbilical cord endothelial cells) [156,350]. In this physiological scenario, the blood/CSF provides a barrier in the choroid plexus for ZIKV invasion to the nervous system, which could regulate virus access to the interior brain and spinal cord surfaces. The choroid plexus is a complex structure to develop a model to study its interplay with ZIKV to characterize ZIKV infection and trafficking into the fetal brain. However, it is conceivable that ZIKV may traffic across the choroid epithelium via transcytosis and exocytosis into the CSF [351]. Indeed, ZIKV-mediated injury patterns around the lateral ventricles (where CSF flows from) have been reported in both humans and non-human primates [352]. Likewise, ZIKV persistence in the placental chorionic villi points to a potential establishment of a viral reservoir in this tissue that sustains maternal viremia throughout pregnancy [16]. Furthermore, the blood/brain barrier (BBB) could mediate in the access of the ZIKV to the outer brain and spinal cord regions, whereas the blood/retinal barrier could regulate ZIKV invasion to the developing ocular tissues [130,349,353].

Overall, at the maternal/fetal interface, different cell types express cell-surface factors that allow viral entry and infection, replication and further transmission to the fetus, such as Axl, Tyro3 and TIM1 [156] that belong to the ZIKV PS receptors of the TAM (i.e., Tyro3 and Axl) and TIM families [168]. Animal models show that ZIKV infection targets neural progenitor cells during pregnancy that results in reduced fetal cerebral cortical surface area [354], which could trigger the sequence of malformations associated with CZS. This ZIKV cytotoxic effect on neural progenitor cells has also been described by infecting cells, neurospheres and/or cerebral organoids structures made of neural stem cells [140,290,355]. In non-human primates, ZIKV infection has been associated with reduced NSCs proliferation across the subventricular zone, compromising neurogenic output [356,357]. It has also been reported that the multi-ciliated ependymal epithelium, a ventricular-subventricular stem cell niche during human brain development [358], could be injured by ZIKV thus impairing CSF transport and outflow through the dura, which might contribute to ventriculomegaly, both in non-human primates and humans [70]. Different data point to the possibility that ZIKV could pass through the BBB to access outer brain cells at later stages of fetal development, since ZIKV infection of this tissue is well correlated with the presence of posterior cortical/subcortical calcifications and gliosis, and polymicrogyria (i.e., late migration defects) [359,360]. These cytopathic events could be responsible for or contribute to motor, vision, auditory, arthrogryposis, and dystonic disorders [359,361]. Once ZIKV crosses the BBB to access outer parenchyma, the virus could invade outer NSCs, specialized neurogenic daughter cells (i.e., intermediate progenitors (IPs)), neurons, as well as astrocytes and microglia [304,305,362,363]. This infection process leads to a loss of progenitors and neuroinflammation together with immune activation and cytokine release that abrogate neurogenesis (reviewed in [351]). Additionally, innate antiviral pathways are negatively altered by ZIKV infection in early pregnancy and, particularly, when congenital microcephaly is relevant [156]. There is an important body of knowledge about the harmful effects exerted by this flavivirus on the developing brain. However, the understanding of the mechanisms underlying the consequences of ZIKV infection on mature CNS requires more research. As introduced in the present review, a broad spectrum of neurological disorders has been reported in adult patients infected by ZIKV [36], detecting ZIKV in CSF and brains of adult patients [24,36,37,39,41]. In animal models, ZIKV has also been found in CSF of infected monkeys, persisting for weeks in this cerebral fluid, even after viral clearance of the rest of the organism [151], whereas ZIKV infects neural progenitors, affecting their proliferation in the adult brain of mice [364]. Additionally, in adult infected mice, genomic RNA of ZIKV has been detected in the frontal cortex and hippocampus, and observing synapsis impairment by the virus, thereby indicating that ZIKV could target memory-related brain regions [365]. These negative effects of ZIKV on synapsis and memory in mice appear to be related to the upregulation of the TNF-α signaling, subsequent microglial activation and inflammation together with significant increase of C1q/C3 proteins of the complement system. In this work, data demonstrate that ZIKV replicates in ex vivo tissues isolated from human adult brain, and targets mature neurons [365]. Altogether these data suggest that ZIKV could infect and persist in adult CNS, accounting for the long-term neurodegenerative impact of ZIKV infection in the brain of adult patients.

The observation of conjunctivitis and uveitis events together with other ocular injuries associated with ZIKV infection in children and adults, such as optic nerve hypoplasia, coloboma, foveal reflex loss, vascular defects, and gliosis and optic nerve cupping/neuritis related to inflammation at later fetal stages are indicative that the virus is able to infect and injure the eye and the retina [349,366,367], and that it may cross the nervous system protective blood/retinal barrier.

Taken together these data are key to understanding the biological processes, as well as the cellular and molecular mechanisms that could favor ZIKV maternal/fetal transmission and trafficking into the brain, as reported in ZIKV infected monocytes that present enhancing adhesion and transmigration capacities favoring viral dissemination to neural cells [368], in order to elicit fetal protection and develop antiviral therapeutics or vaccines [128,129,369,370,371,372].

## 3. Conclusions

ZIKV persistence and pathogenesis involve a complex immune-evasion strategy to facilitate ZIKV trafficking, infection and spread through different cell types and tissues to finally cross protective barriers to the immune-privileged fetus, affecting the development of the fetal brain and eyes.

ZIKV uses its own viral proteins, gRNA and sfRNA elements to evade antiviral immunity, particularly the anti-ZIKV IFN response and associated signals and factors, as mentioned above. Hence, multiple NS proteins of ZIKV negatively modulate the antiviral response at various levels by inhibiting type I IFN production and the expression of downstream ISGs, acting on the IFN-associated cGAS-STING pathway, targeting with TBK1 and therefore impairing IRF3 promoters, or modulating RIG-I- and MDA5-directed type I IFN induction, as well as different steps of the antiviral type I IFN system against RNA viruses such as ZIKV. The NS proteins could cooperate to overcome the INF antiviral functions, limiting immune-protection and the understanding of ZIKV immune-evasion and pathogenesis. Moreover, gRNA-associated modifications that function as an antagonist of the innate type-I IFN response, and sfRNA stability against XRN-1, 5′-3′ exoribonuclease, which help to efficiently suppress RIG-I- and MDA-5-mediated IFN have been identified as an important proviral pathway, abrogating IFN neutralization of ZIKV infection during the early-phase of the viral life cycle. Moreover, ZIKV infection modifies the miRNA landscape of host cells in order to evade innate and adaptive immune responses and promote viral replication and survive. Although the involvement of viral antigen specific CD4^+^ T cells in the control of ZIKV infection and disease is still controversial, the CD8^+^ T cell response is associated with the control of the ZIKV infection and pathogenesis. This protective action has been more clearly demonstrated by cross-immune reactions of human DENV-elicited CD8^+^ T cells [326] which react against ZIKV-NS (i.e., mainly recognizing NS3 protein [329,330]). Anti-ZIKV vaccines are therefore focused on generating protective CD8^+^ and CD4^+^ T cells in order to control ZIKV infection and promote virus clearance, thereby avoiding harmful ADE effects with potential reinfection events by ZIKV or other flaviviruses [121,128,373,374,375].

In order to reach the human fetus, ZIKV must cross the placenta barrier which develops within days of conception and is indispensable for pregnancy [376,377]. Furthermore, fetal damage by ZIKV infection can be observed after the first trimester, and persistence infection, late during pregnancy, results in fetal disease or adverse pregnancy outcomes [13,378].

Recently, as a result of the global impact of ZIKV and its teratogen effects, such as microcephaly in newborns to infected mothers along with neurological abnormalities and GBS, and comparison with previous congenital pathogens such as *Toxoplasma gondii*, other (e.g., syphilis, human immunodeficiency virus (HIV) or parvovirus B19), rubella virus, cytomegalovirus (CMV) and herpes simplex virus (HSV) (i.e., all of which are grouped under the term of TORCH pathogens) [379]. ZIKV has been proposed as a new TORCH pathogen but more complex in the associated pathogenesis [379]. ZIKV-induced congenital microcephaly abnormality alarmed the whole of the planet [19] together with other developmental abnormalities that determine the CZS (i.e., severe microcephaly with partially collapsed skull, thin cerebral cortices with subcortical calcifications, macular scarring and focal pigmentary retinal mottling, congenital contractures and marked early hypertonia) [380,381], and placental insufficiency and fetal loss [71]. ZIKV is able to infect, traffic and spread through several cells and tissues during fetal differentiation, just after overcoming/crossing several protective barriers such as those mentioned above. ZIKV is able to infect eye and brain immune-privileged organs, avoiding innate IFN defenses and provoking neuroinflammation and cytokine release that halt neurogenesis [351], thereby favoring viral spread and tissue damage.

Therefore, understanding how ZIKV interacts with the INF system and influences the outcome of infection in barrier tissues such as the placenta, eyes and brain during pregnancy may help in the development of therapeutic strategies to clear ZIKV from the organism, based on the action mechanism of the IFN system, and keeping immune-privileged organs safe during both fetal development and in adults.

Moreover, determining the molecular interplay between the ZIKV genome and viral proteins with the IFN and T cell responses and antiviral signals is also crucial to improving vaccine strategies based on mutant live-attenuated viral candidates that need to induce adaptive cellular and humoral immune protective responses, also helping type-I IFN antiviral responses [128,129,169,382].

## Figures and Tables

**Figure 1 vaccines-09-00294-f001:**
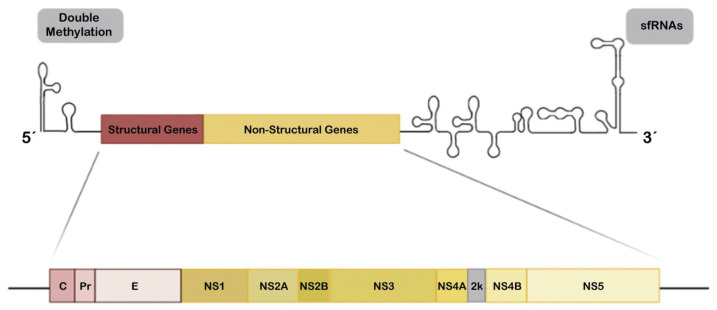
Schematic overview of the Zika virus (ZIKV) genome. The ZIKV genome is a single-stranded positive-sense RNA (+ssRNA) of approximately 11 Kb and codes a polyprotein which is cleaved in the endoplasmic reticulum (ER) lumen by the host and/or viral proteases to release three structural proteins (C, PrM and E; red boxes) and seven non-structural proteins (NS1, NS2A, NS2B, NS3, NS4A, NS4B and NS5; yellow boxes). The 5′ untranslated region (UTR) is susceptible for specific double methylation (represented in grey box at left) by the viral C-terminal methyltransferase domain of NS5 to behave like messenger RNA (mRNA) and hijacks host factors for translation and for masking the viral RNA to prevent cellular recognition and degradation. UTR loops at 3′ end are subgenomic flavivirus RNAs (sfRNAs) (shown in the grey box on the right), that play a role in the immune response evasion (see Section 2.4 in the main text).

**Figure 2 vaccines-09-00294-f002:**
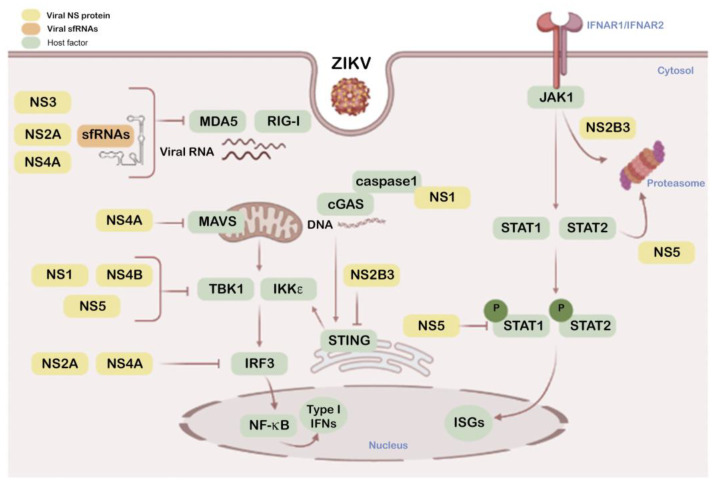
Schematic overview of ZIKV NS proteins and sfRNAs as modulators of antiviral interferon (IFN) and associated signals. This diagram shows the interactions of cellular factors (green boxes) involved in the IFN pathway with the NS viral proteins (yellow boxes) and sfRNAs (sfRNA1 and 2) (orange box) that play a role in this IFN pathway. Color code inbox is indicated top left.

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
