# Peer review of "Zika Virus Pathogenesis: A Battle for Immune Evasion"

_vaccines, 2021, doi:10.3390/vaccines9030294_

Round 1
Reviewer 1 Report
Estimated Authors,
Estimated Editors,
thank you for the opportunity to review this outstanding review on the Zika virus pathogenesis, and particularly on the relationships between ZIKV and immune system.
In my opinion, Authors have collected and properly reported all the available evidence on the topic they have identified. Therefore, I've only the following, minor, recommendations:
1) modify the main title in order to define the present study as a "narrative review"
2) include as a table the summary research strategy you applied in order to collected the evidences you summarized.
Again, I congratulate with you for the very high quality of this paper.
Author Response
We would like to thank you for your kind words and evaluation of our manuscript entitled “Zika virus pathogenesis: a battle for immune evasion” by Judith Estévez-Herrera and co-workers.
We have highlighted all modifications in red color texts.
We have attempted to provide a deep interpretation and critique of the topics addressed in this review, considering data with a historical perspective in the wide range of studies on the different ZIKV outbreaks, and analyzing relevant original works, as well as the main comprehensive reviews that have shed light on the impact of ZIKV on immune functions, and how the virus evades immune functions, replicates and persists, causing the complex ZIKV pathology. This task prompted us to perform a rigorous conceptual analysis of re-search works that have challenged arguments, identified and resolved contra-dictions that are associated with the study of the ZIKV infection, the complex mechanisms for immune evasion and its pathology, in order to facilitate the understanding of these research works and their impact in the battle against ZIKV and associated complex disease.
We are pleased to include these kinds of considerations in the main title page, if the Editors of Vaccines and of the associated Special Issue of "Zika Virus and Immune Response” consider these categorizations as suitable for our review.
In this regard and as suggested by Reviewer#1, we have included a Table (Table 1) summarizing the research strategy used to prepare the present review. Please, find Table 1 on page 3, before section 2. “Mechanisms of ZIKV immune evasion”.
Reviewer 2 Report
Major comments:
I would like to thank the authors for submitting this review. The review is very detailed in terms of the innate immune response and how the Zika virus circumvents it.
- However, one the major concerns is that the authors have not described about what role adaptive immune response plays. This is extremely important as authors have mentioned that one of the reasons this interaction is so important because it would be helpful for development of vaccines. There are several studies looking at CD8 and CD4 responses in the context of Zika virus.
- The authors can also discuss the role T cell in cross protection and in the context of re-infection. Which epitope do the specific T cells target during infection and which of them provide an effective immune response? Can the virus escape these immune mechanisms? Grubor-Bauk et al, 2019 (Science), published a NS1 DNA vaccination paper (in mice). The authors might comment on whether such vaccine strategies be effective in humans.
- The authors might also want to discuss role/potential role of microRNA in the immune escape of Zika virus.
Minor comments:
- In the introduction the authors have mentioned about the complications seen in fetuses. The authors might also want to focus on severe disease/complications seen in adults. This is needed as later before section 2.1 authors mention about interaction of Zika virus with immune system to describe severe disease.
- Section 2.1 – description about the IFN pathway is very well written.
- Last paragraph of section 2.2 is hard to read.
- Figure 2 – A darker color for viral proteins would make them more prominent. Authors might also add a legend box in the figure describing colors of viral vs host proteins.
Author Response
We would like to thank you for your kind words and evaluation of our manuscript entitled “Zika virus pathogenesis: a battle for immune evasion” by Judith Estévez-Herrera and co-workers.
We have English edited the text of the manuscript, as suggested by Reviewer#2.
We have highlighted all modifications in red color texts.
In major comments:
We would like to thank Reviewer#2 for the kind comment about how we have addressed innate immune response in the context of ZIKV infection.
As requested by Reviewer#2 concerning T cell-mediated adaptive immunity, cross-reactivity, reinfection and vaccines, we have included a new section, pages 11-13, entitled “2.6. ZIKV infection and adaptive immune responses understanding cross-immune protection and vaccine challenges.”.
We have also added a short discussion about T cell immune responses in “3. Conclusions” section, page 15, as follows:
“Moreover, ZIKV infection modifies the miRNA landscape of host cells in order to evade innate and adaptive immune responses and promote viral replication and survive. Although the involvement of viral antigen specific CD4+ T cells in the control of ZIKV infection and disease is still controversial, the CD8+ T cell response is associated with the control of the ZIKV infection and pathogenesis. This protective action has been more clearly demonstrated by cross-immune reactions of human DENV-elicited CD8+ T cells [330] which react against ZIKV-NS (i.e., mainly recognizing NS3 protein [333,334]). Anti-ZIKV vaccines are therefore focused on generating protective CD8+ and CD4+ T cells in order to control ZIKV infection and promote virus clearance, thereby avoiding harmful ADE effects with potential reinfection events by ZIKV or other flaviviruses [124,131,378-380].”.
Considering the remark made by Reviewer#2 about the potential role of microRNA in the immune scape of ZIKV infection, we have included a new section, pages 9-10, entitled 2.5. “MicroRNA, ZIKV infection, immunity and pathogenicity”, where we have also discussed the consequences for pathogenesis.
In minor comments:
We agree with Reviewer#2 concerning neuro-damage caused by ZIKV infection in adults. We have discussed this issue in two parts of the review, as follows:
- In the introduction section, page 2:
“ZIKV has been found in the cerebrospinal fluid (CSF) and brain of adults infected by the virus who manifested neurological disorders [24,36-39]. This flavivirus causes harmful effects in the adult brain, such as GBS [36,38,40-42], encephalitis [36,41-43], meningoencephalitis [24,44], acute myelitis [36,42,45] and encephalomyelitis [36,39,41,46,47], among sensory polyneuropathy [48] and other neurological complications [49,50].”.
- In section 2.7, page 14:
“There is an important body of knowledge about the harmful effects exerted by this flavivirus on the developing brain. However, the understanding of the mechanisms underlying the consequences of ZIKV infection on mature CNS requires more research. As introduced in the present review, a broad spectrum of neurological disorders has been reported in adult patients infected by ZIKV [36], detecting ZIKV in CSF and brains of adult patients [24,36,37,39,41]. In animal models, ZIKV has also been found in CSF of infected monkeys, persisting for weeks in this cerebral fluid, even after viral clearance of the rest of the organism [154], whereas ZIKV infects neural progenitors, affecting their proliferation in the adult brain of mice [368]. Additionally, in adult infected mice, genomic RNA of ZIKV has been detected in the frontal cortex and hippocampus, and observing synapsis impairment by the virus, thereby indicating that ZIKV could target memory-related brain regions [369]. These negative effects of ZIKV on synapsis and memory in mice appear to be related to the upregulation of the TNF-a signaling, subsequent microglial activation and inflammation together with significant increase of C1q/C3 proteins of the complement system. In this work, data demonstrate that ZIKV replicates in ex vivo tissues isolated from human adult brain, and targets mature neurons [369]. Altogether these data suggest that ZIKV could infect and persist in adult CNS, accounting for the long-term neurodegenerative impact of ZIKV infection in the brain of adult patients.”.
We would like to thank Reviewer#2 for the kind comment regarding the IFN 2.1 section.
We have attempted to rewrite the last paragraph in section 2.2., in order to facilitate its reading, as requested by Reviewer#2.
As requested by Reviewer#2, in Figure 2, we have only indicated in bold all the viral NS proteins and sfRNAs. In the table legend, the color code for NS, sfRNAS and host factors is indicated in parentheses. We have added a color code inbox in the top left of the figure.
In different parts of the manuscript, we have included words and concepts related to all the above issues as requested by Reviewer#2.
Reviewer 3 Report
The manuscript of Estevez-Herrera and co-authors represents an in-depth review of the mechanisms employed by Zika virus for evasion of the human immune response. This review is well-written and current. It characterizes thoroughly all expects of ZIKV-host interactions, involving protein and RNA-based mechanisms employed by virus for inhibition of immune response. I am confident that it will be of interest for a wide community of virologists and immunologist. However it has room for improvement as reflected in the following minor suggestions:
- The manuscript requires minor stile editing. Please, reduce the use of “in fact” and “indeed”.
- I would rather disagree with a statement on P3 (section 2.1) that “every cell under attack has a viral guard”. IFNs and other cytokines trigger antiviral response in uninfected cells too, making them resistant to infection. Uninfected cells don’t have a viral guard.
- In section 2.4 authors indicate that sfRNA seems to supress IFN induction, which was demonstrated using reporter system. However, the recent study (PMID: 32581095) demonstrated that production of IFNs is not inhibited by ZIKV sfRNA in real infection. I believe it will be fare to unknowledge the existence of the alternative opinions regarding the ability of ZIKV sfRNA to inhibit PRP-signalling and IFN induction.
- I would suggest changing the title of section 2.4 into “Viral genomic and subgenomic RNAs inhibit IFN signalling” because currently it reads like all this functions are executed by genomic RNA only. I might be also worth mentioning that ZIKV mutant deficient fully deficient in sfRNA was shown to be unviable in vertebrate and insect cells, which stresses on the importance of sfRNA in immune evasion (PMID: 32371874).
Author Response
We would like to thank you for your kind words and evaluation of our manuscript entitled “Zika virus pathogenesis: a battle for immune evasion” by Judith Estévez-Herrera and co-workers.
We have thoroughly English edited the text, as requested by Reviewer#3.
We have highlighted all modifications in red color texts.
Concerning the remark raised by Reviewer#3 with respect to the following sentence “Every cell under attack has a viral guard.” (former Page 3, section 2.1; now on page 4, section 2.1), we aimed to address the processes triggered only in infected cells, from the beginning of the section. Therefore, we presented this paragraph after the said sentence:
“ZIKV responds to the assault by the infected cell by 1) mounting multiple camouflage strategies evading free genomic RNA recognition in cytosol, 2) avoiding the activation of multiple interactions of IFN cascades and finally 3) turning off the signaling that serves as a warning signal of infection.”
In this matter, we have modified the sentence (now on page 4, section 2.1), as follows:
“Every cell under attack has a viral guard or is protected by the immune alarm that mounts a viral guard, establishing an antiviral state making them resistant to infection.”.
Following on with this issue, we agree with Reviewer#3 about the comment that IFN mediates paracrine cell protection in neighbor non-infected cells, preparing them to be less permissive for viral infection. This idea is now exposed in a following paragraph (red color text) on page 6, section 2.1, as follows:
“Furthermore, paracrine IFN signaling in non-infected cells renders these cells refractory to viral infection. All type I IFNs are able to signal through the type-I IFNAR that is composed of two heterodimeric subunits (IFNAR1 and IFNAR2) which are generally widely distributed throughout the body (reviewed in [201,202]). This canonical type I IFN signal induces a plethora of ISREs which drive ISGs expression to establish a cellular antiviral state ([203-206]) (Figure 2). The protective role of IFN-I against ZIKV has been demonstrated in IFN-I signaling-deficient mice which are highly susceptible to viral infection [207]. Type I and II IFN-pretreated primary skin fibroblasts showed a non-permissive status against ZIKV, observing a strong and dose-dependent inhibition of viral replication [183]. In a different work, IFN-β shows moderated anti-ZIKV activity [208]. However, elevated secretion of IFN-β, detected in human lung epithelial cells during ZIKV infection appears to be responsible for preventing virus-mediated cell death by apoptosis [208]. Concerning the protective role of ISGs against ZIKV infection, it has been reported that placental cells could resist viral infection due to actions of IFN-λ (type III IFN) [209]. Likewise, it seems that small membrane-associated IFN-inducible trans-membrane proteins (IFITMs), particularly IFITM1 and 3, could inhibit ZIKV infection early in the viral life cycle [210]. In this work, results point to the possibility that IFITM3 could also prevent cell death mediated by ZIKV infection [210].”.
As requested by Reviewer#3, “in section 2.4”, we have discussed the referred to “PMID: 32581095 (Pallarés, H-M., et al. (2020))” work that indicates that ZIKV sfRNAs could not alter IFN production during active infection. We entirely agree with Reviewer#3 regarding published data by Pallarés, H-M., et al. (2020). In this work, the authors characterize the molecular determinants for ZIKV sfRNA generation in the two natural hosts, human cells and mosquitoes. Moreover, we have discussed data from reference “PMID: 32371874 (Slonchak. A., et al. (2020)”, concerning the importance of sfRNAs for ZIKV infection in vertebrates and invertebrates.
We have discussed these data on page 9, section 2.4, as follows:
“In this regard but without targeting IFN production, recent studies using wild-type ZIKV point to the importance of sfRNAs for viral infection in both human cells and mosquitoes [268]. ZIKV-sfRNAs do not negatively affect IFN levels, since high IFN and proinflammatory cytokines levels were induced after infection of human cells with wild-type virus [268]. However, IFN levels were strongly diminished in cells infected by a recombinant ZIKV unable to generate sfRNAs [268]. Of note, efficient viral clearance was observed in cells infected with sfRNA-deficient ZIKV. Further studies are needed to clarify this apparent contradictory observation concerning the absence of sfRNA-mediated regulation of IFN immunity. However, these data correlated well with the strong induction of ISGs which occurred in these infected cells, thereby suggesting that the deficiency in sfRNA is associated with an uncontrolled IFN signaling which strongly activates ISGs to further clear ZIKV [268]. Therefore, sfRNAs are required for ZIKV propagation in human cells. Furthermore, results obtained with recombinant ZIKV indicated that the absence of sfRNAs results in inefficient viral infection and transmission in Aedes aegypti mosquitoes [268]. Similarly, a recent study has demonstrated the importance of the sfRNAs for productive infection and virus transmission in mosquitos by suppressing ZIKV infection-mediated apoptosis and assuring the ability of the virus to disseminate and reach saliva in the invertebrate vector [269].”.
As suggested by Reviwer#3, we have modified the subheading in section 2.4, as follows:
“2.4. Viral genomic and subgenomic RNAs inhibit IFN signaling.”
Round 2
Reviewer 2 Report
I would like to thank the authors for addressing the major comments on miRNA and Adaptive immune system.
Author Response
Friday, March 12, 2021
Dear Reviewer#2 of Vaccines,
Answer to Reviewer#2:
- The Reviewer#2 have stated that “I would like to thank the authors for addressing the major comments on miRNA and Adaptive immune system.”.
We would like to thank you for your kind words and evaluation of our manuscript that has improved this work.
We would like to thank the Academic Editor and the three reviewers for helping us to improve this piece of work, and hopefully it will now be considered suitable for publication in Vaccines.
We certify that this manuscript is original and that it is not under consideration by another journal, nor has the data been previously published.
We declare that there are no conflicts of interest in relation to the submitted work, all the authors concur with the submission of this research work, and we accept responsibility for this material. The other co-authors agree to the manuscript’s submission and that I will act as corresponding author.
Looking forward to hearing from you soon.
Yours sincerely,
Agustín Valenzuela-Fernández, PhD
Laboratorio de Inmunología Celular y Viral
Universidad de La Laguna (ULL)
Tenerife, Spain.
E-mail: avalenzu@ull.edu.es
____